# Effects of Low-Level Laser Therapy and Bracket Systems on Root Resorption during Orthodontic Treatment: A Randomized Clinical Trial

**DOI:** 10.3390/healthcare11060864

**Published:** 2023-03-15

**Authors:** Fazal Shahid, Shifat A Nowrin, Mohammad Khursheed Alam, Mohd Fadhli Khamis, Adam Husein, Norma Ab Rahman

**Affiliations:** 1Orthodontic Department, Shifa College of Dentistry, Shifa Tameer-e-Millat University, Islamabad 44000, Pakistan; 2Orthodontics, Preventive Dentistry Department, College of Dentistry, Jouf University, Sakaka 72345, Saudi Arabia; 3Center for Transdisciplinary Research (CFTR), Saveetha Institute of Medical and Technical Sciences, Saveetha Dental College, Saveetha University, Chennai 602105, India; 4Department of Public Health, Faculty of Allied Health Sciences, Daffodil lnternational University, Dhaka 1216, Bangladesh; 5Forensic Dentistry, School of Dental Sciences, Universiti Sains Malaysia, Kubang Kerian 16150, Kelantan, Malaysia; 6Department of Preventive and Restorative Dentistry, College of Dental Medicine, University of Sharjah, Sharjah 27272, United Arab Emirates; 7Prosthodontic Unit, School of Dental Sciences, Universiti Sains Malaysia, Kubang Kerian 16150, Kelantan, Malaysia; 8Orthodontic Unit, School of Dental Sciences, Universiti Sains Malaysia, Kubang Kerian 16150, Kelantan, Malaysia

**Keywords:** laser therapy, bracket systems, root resorption

## Abstract

This study aimed to assess the outcomes of low-level laser therapy (LLLT) with the conventional bracket (CB) and self-ligating (SL) bracket systems on root resorption (RR) during orthodontic treatment. A total of 32 patients were included in this randomized clinical trial. All the patients were randomly divided into four individual groups (SLL: self-ligating laser, CBL: conventional bracket laser, SLNL: self-ligating non-laser, CBNL: conventional bracket non-laser). RR was measured from the cone-beam computed tomography (CBCT) radiographs which were taken at two stages of the orthodontic treatment: pre-treatment (T1) and after leveling and alignment stage (T2). Wilcoxon rank test for the comparison was conducted to compare the RR at T1 and T2 stages within each group and showed a significant difference (*p* < 0.05) for various variables. Mann Whitney test compared the RR in laser and non-laser groups irrespective of the bracket systems and exhibited no significant differences except the left lateral incisor. Moreover, CB and SL groups showed no significant difference in RR among any tooth. Kruskal Wallis test was performed to compare the RR among all groups which presented no significant differences. LLLT and bracket systems have no consequences on RR until the leveling and alignment stage of orthodontic treatment.

## 1. Introduction

Root resorption (RR) due to orthodontic treatment is a very prevalent hurdle [1]. During fixed orthodontic treatment multiple factors influence the RR. The factor might be scaled of orthodontic force based on the duration of treatment [2,3]. Similarly, the method of force application, tooth movement direction, the extent of radicular tooth movement, and memoir of trauma are related to RR [4]. Likewise, RR can be predisposed by different systemic diseases, for example, scleroderma, Papillon-Lefevre syndrome hypoparathyroidism, and thalassemia major [4]. During orthodontic treatment, about 4% of subjects reach 3 mm RR and 5% could experience 5 mm or more [1,5].

RR is a common phenomenon which observed in most of the cases after finishing the orthodontic treatment. Various mechanical factors might involve in RR during orthodontic treatment such as excessive force, intrusion, extrusion and root tipping. These mechanical factors usually applied during the retraction stage or finishing stage of orthodontic treatment. Perhaps, at the initial stage or the levelling and alignment stage of orthodontic treatment, a mechanism that leads to enhancing the RR such as the application of heavy force is not much applicable. As per literature search, none of the previous study assessed the RR in the initial stage of the orthodontic treatment. Therefore, it is necessary to assess whether the root resorption occur in initial stage and at what extend this phenomenon observe until the initial stage.

RR could be assessed with the routine radiographs taken during the orthodontic treatment. RR could be assessed with the routine radiographs taken during the orthodontic treatment. Orthodontic routine radiographs such as orthopantomogram (OPG), lateral cephalogram or posterior-anterior (PA) are constructed with two-dimensional (2D) images, whereas cone-beam computed tomography (CBCT) produces three-dimensional (3D) insignificant images which could not be viewed with the routine 2D radiographs [6]. In addition, 3D images software could construct 3D images in any specific area which supports orthodontists to foresee the area of interest. In recent era, CBCT considered as a golden standard for obtaining any defects in the hard tissue specifically in the maxillofacial region. A CBCT scan plays a significant role in the evaluation of diagnostic images. High-resolution 3D images of patient’s anatomy eradicate any guesswork and help clinicians to make informed, educated and accurate decisions regarding treatment [7].

Laser therapy has been used in orthodontic treatment for many years [8]. There are different types of lasers exist and have been used in orthodontic treatment. Due to the nature of stable temperature, low-level laser therapy (LLLT) is identified as a ‘cold laser’ and does not intensify the temperature in tissues unlike other types of lasers that were utilized in thermal coagulation or cutting of the tissues [9]. Furthermore, cellular assimilation of the laser light by the target tissues mitochondria initiates implying cascades [10] and promoting vascularization, epithelization, and collagen synthesis [11]. LLLT application influences the outcome of orthodontic therapy as it might affect the tooth movement due to its anti-inflammatory, analgesic, and bio-stimulatory influence on the tooth and periodontium [4,12]. Suzuki et al. (2018) presented a less crater on the tooth root in rats treated with LLLT [12]. On the other hand, Marquezan et al. (2013) found evidence of RR on the eighth day of orthodontic tooth movement (OTM) with the application of LLLT [13]. Due to the contrasting outcomes, this study intended to assess the effectiveness of LLLT on RR via cone-beam computed tomography (CBCT) along with the conventional bracket (CB) and self-ligating (SL) bracket systems until the leveling and alignment stage of the orthodontic treatment. The null hypothesis of the current study is there is no significant difference in the effect of LLLT and non-LLLT groups in relation to root resorption for extraction case management with the CB and SL bracket systems until the levelling and alignment stage of orthodontic treatment via 3D CBCT.

## 2. Materials and Methods

This is an in-vivo study, designed as a randomized clinical trial for patients under treatment with fixed orthodontic brackets in the Orthodontic Specialist Clinic of Hospital, Universiti Sains Malaysia from 2016 to 2019. The reference population was orthodontic patients with moderate crowding requiring the extraction of the first premolars. The human research ethical committee (HREC) of Universiti Sains Malaysia approved this study with the reference number (USM/JEPeM/17070339). The following inclusion criteria were followed: age 18–35 years, no history of past orthodontic treatment, the eruption of all teeth except 3rd molar, moderate crowding (4–10 mm), no root resorption, and CBCT images. Patients with interproximal caries or restoration, missing or supernumerary anterior teeth, abnormal morphology of teeth, patients with medical problems, parafunctional habits, craniofacial malformation, periodontal disease and patients who missed their scheduled appointment were excluded from this study. This randomized clinical trial followed the CONSORT guidelines (Figure 1).

Sample size calculation:

Power and sample size calculation software (version 3.1.6) [14] was used to determine the sample size calculation of this study. The type I error of 0.05, power of 80%, the standard deviation of 0.99 and mean difference of 1 estimated a total sample size of 32 for this study.

Variables:

The independent variables for this study were, LLLT and Non-LLLT groups, conventional brackets and self-ligating brackets. The dependent variable measured via the CBCT was RR which was recorded as blunting or shortening of the root apex.

Apparatuses to conduct during the research:

Passive self-ligating MBT (McLaughlin Bennett Trevisi) brackets (smart clip, 3M Unitek, Monrovia, CA, USA) and pre-adjusted edgewise conventional MBT brackets (Ortho Organizers, Carlsbad, CA, USA) with a 0.022-inch slot were used as a bracket of choice. Transbond XT light-cure bonding kit (3M Unitek, Monrovia, CA, USA) was used for the bonding in patients’ arches. Both 1st and 2nd molars were bonded with the buccal tubes (3M Unitek, Monrovia, CA, USA). The archwires sequence used for both brackets systems were 0.014, 0.016, 0.017 × 0.025, and 0.019 × 0.025-inch heat-activated nickel-titanium (NiTi) archwire of 3M Unitek (Monrovia, CA, USA). A Ga-Al-As diode laser (iLas; Biolase, Irvine, CA, USA) was used for laser application with a 940 nm in a continuous [15], self-ligating pliers, CBCT Planmeca Promax 3D machine with Planmeca RomexisTM software (Planmeca, Helsinki, Finland).

Data collection procedure:

A total of 170 patients were assessed for selection in the current research. Based on the inclusion criteria, declined participation, and other reasons a total of 138 patients were excluded. A total of 32 patients were selected according to the randomization plan. All the patients who were diagnosed with mild crowding (4–10 mm) [16] which required bilateral extraction of either first or second premolars in both arches were selected for this study. None of the patients required special anchorage management irrespective of the classification of the malocclusion. Dental casts and routine radiographs for the orthodontic treatment were collected from the patients and analyzed. All the treatment procedure was explained to the patients and those who agreed to participate in this study signed the informed consent form. CBCT radiographs were obtained at two stages of the treatment: before starting the treatment (T1) and after finishing the leveling and alignment of the treatment (T2). CBCT images were constructed with the exposure of 84 kVp, 8 mA, 320-μm voxel resolution, and FOV 16 cm. All the patients were randomly and equally divided into four groups: SLL: self-ligating laser, CBL: conventional bracket laser, SLNL: self-ligating non-laser, and CBNL: conventional bracket non-laser. During the follow-up, none of the patients were discontinued from each group.

LLLT was applied in all anterior teeth in CBL and SLL groups. A gallium-Aluminum-Arsenide (Ga-Al-As) diode laser (iLas; Biolase, Irvine, CA, USA) with a continuous 940 nm wavelength, 100 mW laser output with 7.51 J/cm^2^ energy density per tooth was used in the current study [17]. Laser applied in a total of five points (mesio-apical area and disto-apical area of root, middle of the root, mesio-cervical and disto cervical area of the root) for six seconds in each tooth on every visit until the levelling and alignment stage of the orthodontic treatment. An isolated room in the clinic was used for the LLLT application with protective eyewear for the operator, patients and dental assistant.

Root resorption assessment via CBCT:

RR of the anterior teeth was evaluated via the CBCT images with Planmeca Romexis^TM^ software 2.3.1.R (Helsinki, Finland). The tooth length of each anterior tooth was measured by dividing the tooth into two portions with cementoenamel junction (CEJ): crown (C) and root (R) (Figure 2). Apex to CEJ is considered as root length and CEJ to the tip of the cusp is considered as crown length. Crown length should remain consistent over the period of observation; therefore, a correction factor (CF) was calculated [15]. RR was measured from the CBCT images by using the following formula:

RR = R1 − (R2 × CF); CF = C1/C2 [18]

[R1 = root length measured at T1 stage (mm), R2 = root length measured at T2 stage (mm), C1 = crown length measured at T1 stage (mm), C2 = crown length measured at T2 stage (mm)] (Figure 3).

**Figure 2 healthcare-11-00864-f002:**
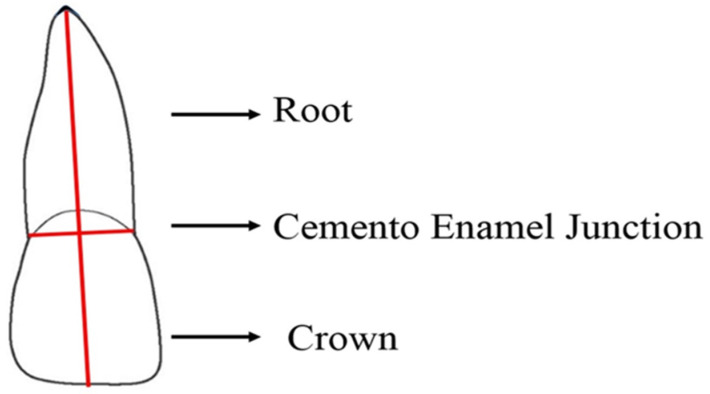
Diagrammatical presentation for the measurements of the tooth root and crown length.

**Figure 3 healthcare-11-00864-f003:**
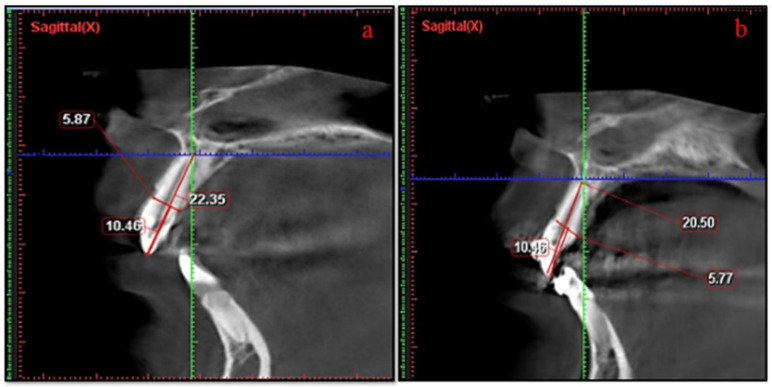
Measurements of root resorption (**a**) Sagittal view of CBCT showing the pre-measurement of root length (**b**) Sagittal of CBCT showing the post-measurement of root length.

Statistical analysis:

All statistical analyses were performed using the SPSS software version 24 (IBM, Chicago, IL, USA). The Shapiro-Wilk test was used to evaluate the normal distribution of the data. Intra-class and inter-class reliability was assessed using the intra-class correlation coefficient (ICC) test. The overall tooth length changes in each group at T1 and T2 stages were observed with the Wilcoxon rank test. Mann Whitney test was conducted to compare the changes between LLLT and non-LLLT groups regardless of brackets, and CB and SL bracket groups regardless of LLLT application, respectively. The changes among all four groups were observed with the Kruskal-Wallis’s test.

## 3. Results

A total of 32 patients with a mean age of 22.41 (4.18) years were participated in this study. The Shapiro-Wilk test showed that data were not normally distributed; therefore, non-parametric tests were performed for all the analyses. A total of eight samples (25% of the total sample) were randomly selected for the reliability test. The ICC showed an excellent correlation in both intra-examiner and inter-examiner reliability examinations for all the variables.

All the groups showed significant differences (*p* < 0.05) for most of the teeth length changes from the T1 to T2 stages (Table 1, Table 2, Table 3 and Table 4). Mann Whitney test indicated that only the left lateral incisor showed a significant difference (*p* = 0.029) in RR compared between the LLLT and non-LLLT group, irrespective of the bracket system. However, no other teeth showed any significant differences though the median value for the RR was higher in the non-LLLT group than in the LLLT group (Table 5). Bracket systems exhibited no significant differences in RR regardless of the laser application (Table 6). Moreover, the Kruskal Wallis test also did not show any significant difference in RR among all four groups (Table 7).

## 4. Discussion

The present study aimed to assess the influence of laser application on RR with CB and SL bracket systems until the leveling and alignment stage of the orthodontic treatment. The present study exhibited that all the anterior teeth significantly reduced the tooth length after the T2 stage of the treatment. The range of RR for groups SLL, CBL, SLNL and CBNL were 0.61 mm–1.45 mm, 0.41 mm–1.65 mm, 0.67 mm–2.06 mm and 0.77 mm–2.96 mm, respectively with greater root resorption for CBNL group. Therefore, it might relate to both the bracket system and the LLLT application.

Orthodontic treatment usually consequences RR most of the time. Hence, different treatment modalities were studied in order to diminish this factor [19]. Various factors play an important role in predisposing RR for example root morphology, tooth anatomy, genetic factors and the orthodontic treatment procedure [19,20,21]. Yet, the current study did not focus on root morphology or genetic factor and assessed the root resorption based on only moderate crowding extraction cases. The most used radiograph to determine RR is panoramic radiographs, lateral cephalogram, and peri-apical radiographs using several methods [22,23,24,25]. However, this study measured the RR with CBCT images which produce a high level of reproducibility [26,27,28].

Many studies previously conducted the comparison of CB and SL bracket systems in RR [29,30]. However, only maxillary arch or maxillary incisors were only assessed for the RR in most of the studies. The current study assessed all the anterior teeth in both arches. Most of the studies were retrospective studies except one study was RCT similar to the current study [31]. Moreover, most of the studies used conventional radiographs such as peri-apical or panoramic radiographs whereas CBCT images were used by only one previous study to assess the RR [32]. Despite all additional factors, none of the previous studies showed any significant association between RR and bracket systems which complies with the outcome of the current study.

The current study also assessed the influence of laser application on RR until the leveling and alignment stage of the orthodontic treatment which indicates that only left sided maxillary lateral incisor possessed significant RR with laser application. Other teeth did not show any effects with the LLLT application. Yet, this outcome could not compare to any previous study as to the best of our knowledge no previous study assessed the effects of LLLT on RR until the leveling and alignment stage of the orthodontic treatment. Only two clinical studies were performed on root resorption of the extracted premolars with LLLT application [31,33]. Ng et al. in their clinical study on the maxillary first premolars found that the placebo group showed more RR compared to the laser group. Moreover, LLLT was applied for the first four days and then weekly (days 7, 14, and 21) during the twenty-eight days experimental period with an 808-nm diode laser [31] while the current study applied LLLT every month during the scheduled visits only. However, Khaw et al. studied the influence of LLLT on the healing of root resorption was assessed on extracted teeth using microcomputed tomography and no harm to the teeth roots was observed [33]. However, the study design of the current study is far more different than the Khaw et al. current study focused on the influence of LLLT on RR among all anterior teeth until the leveling and alignment stage of the orthodontic treatment using CBCT images.

The current study assessed the effects of LLLT along with CB and SL bracket systems on root resorption during the levelling and alignment stage of the orthodontic treatment. This study rejected the hypothesis of the current study and exhibited no influence of LLLT or bracket systems when comparing all the groups on root resorption.

The limitation of this current study was RR assessed up to the leveling and alignment stage of the orthodontic treatment only. The comparison of RR in different stages of orthodontic treatment could expand the outcomes. Moreover, the current study only focused on the moderate crowding with extraction cases; however, different classification of malocclusion regardless of extraction could alter the current outcome. Hence, future studies on LLLT and RR with different types of malocclusions with larger sample sizes are recommended.

## 5. Conclusions

In conclusion, though LLLT showed less root resorption, however, it does not significantly affect root resorption during orthodontic treatment except for the left maxillary lateral incisor. Bracket systems also do not affect root resorption. Moreover, while comparing all groups, laser therapy along with different bracket systems has no influence on RR until the leveling and alignment stage of the orthodontic treatment.

## Figures and Tables

**Figure 1 healthcare-11-00864-f001:**
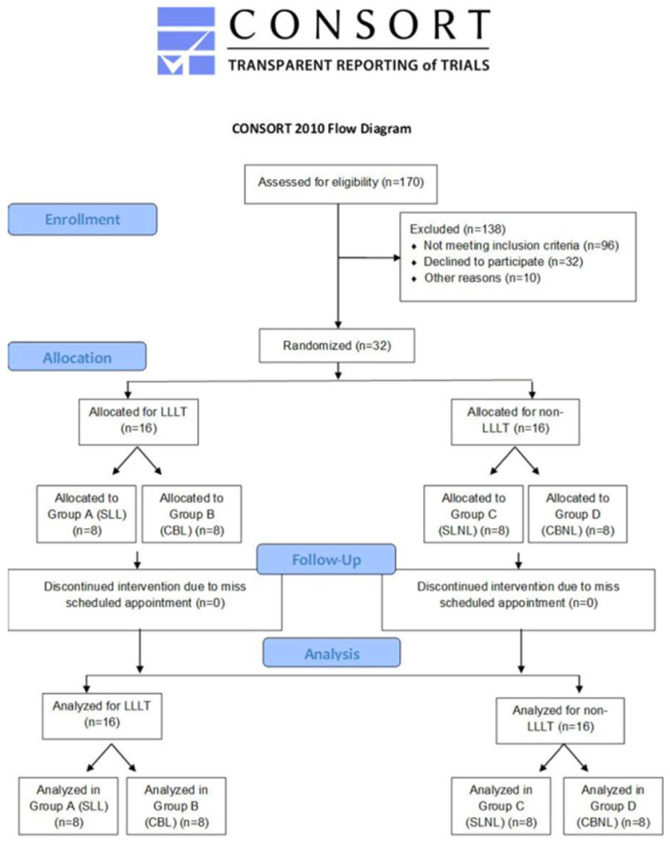
CONSORT guidelines flow diagram.

**Table 1 healthcare-11-00864-t001:** Tooth length (root length and crown length) changes in SLL group.

Variables ^#^	Median (IQR) at T1 (mm)	Median (IQR) at T2 (mm)	Z Statistics	*p*
11	23.36 (3.43)	22.75 (3.46)	−2.52	0.008 *
12	23.11 (1.76)	21.64 (3.55)	−2.52	0.008 *
13	25.47 (5.29)	24.71 (3.95)	−2.52	0.008 *
21	24.09 (1.31)	23.63 (1.77)	−2.52	0.008 *
22	23.26 (2.08)	21.99 (2.23)	−2.52	0.008 *
23	23.28 (4.66)	21.85 (4.76)	−2.52	0.008 *
31	20.37 (1.80)	19.18 (1.63)	−2.52	0.008 *
32	21.17 (2.20)	20.42 (3.33)	−2.36	0.016 *
33	22.58 (6.76)	21.27 (5.48)	−2.52	0.008 *
41	20.64 (0.98)	19.91 (1.75)	−2.52	0.008 *
42	21.76 (3.39)	20.40 (2.02)	−1.96	0.055
43	22.29 (5.14)	22.06 (4.34)	−2.52	0.008 *

Wilcoxon signed rank test; T1, before treatment; T2, at the end of levelling and alignment stage; # FDI notation; * significant difference (*p* < 0.05); mm, millimetre; *p*, *p*-value; IQR, interquartile range.

**Table 2 healthcare-11-00864-t002:** Tooth length (root length and crown length) changes in CBL group.

Variable ^#^	Median (IQR) at T1 (mm)	Median (IQR) at T2 (mm)	Z-Statistics	*p*
11	24.26 (2.03)	23.67 (1.22)	−2.52	0.008 *
12	23.43 (3.03)	21.16 (2.04)	−2.52	0.008 *
13	24.11 (2.60)	23.17 (2.58)	−2.52	0.008 *
21	23.48 (2.42)	23.08 (2.27)	−2.52	0.008 *
22	22.09 (2.30)	21.48 (1.79)	−2.52	0.008 *
23	24.24 (1.83)	23.23 (1.75)	−2.52	0.008 *
31	20.23 (1.82)	19.95 (1.70)	−2.37	0.016 *
32	20.83 (1.34)	19.95 (1.32)	−2.52	0.008 *
33	22.89 (3.57)	22.59 (3.20)	−2.52	0.008 *
41	20.74 (2.88)	19.29 (2.61)	−2.52	0.008 *
42	21.29 (3.12)	20.33 (2.84)	−2.52	0.008 *
43	23.84 (0.68)	22.85 (2.06)	−2.52	0.008 *

Wilcoxon signed rank test; T1, before treatment; T2, At the end of levelling and alignment stage; # FDI notation; SD, standard deviation; * significant difference (*p* < 0.05); *p*, *p*-value; IQR, interquartile range.

**Table 3 healthcare-11-00864-t003:** Tooth length (root length and crown length) changes in SLNL group.

Variable ^#^	Median (IQR) at T1 (mm)	Median (IQR) at T2 (mm)	Z-Statistics	*p*
11	24.83 (2.76)	23.86 (2.05)	−2.52	0.008 *
12	21.39 (3.75)	20.72 (5.15)	−2.52	0.008 *
13	25.19 (2.39)	23.79 (3.29)	−2.52	0.008 *
21	25.20 (2.78)	23.99 (2.60)	−2.52	0.008 *
22	22.99 (1.30)	21.46 (2.74)	−2.52	0.008 *
23	24.94 (3.11)	23.47 (2.33)	−2.52	0.008 *
31	20.69 (1.36)	20.38 (1.55)	−2.52	0.008 *
32	21.82 (1.21)	21.05 (1.17)	−2.52	0.008 *
33	23.98 (3.23)	22.14 (1.84)	−2.52	0.008 *
41	20.70 (2.63)	20.01 (1.35)	−2.52	0.008 *
42	22.72 (1.83)	20.25 (1.95)	−2.52	0.008 *
43	24.05 (4.23)	21.72 (2.83)	−2.52	0.008 *

Wilcoxon signed rank test; T1, before treatment; T2, at the end of levelling and alignment stage; # FDI notation; * significant difference (*p* < 0.05); *p*, *p*-value; IQR, interquartile range.

**Table 4 healthcare-11-00864-t004:** Tooth length (root length and crown length) changes in CBNL group.

Variable ^#^	Median (IQR) at T1 (mm)	Median (IQR) at T2 (mm)	Z-Statistics	*p*
11	24.57 (2.76)	23.86 (2.05)	−2.52	0.008 *
12	21.39 (3.75)	20.72 (5.15)	−2.52	0.008 *
13	25.19 (2.39)	23.79 (3.29)	−2.52	0.008 *
21	25.20 (2.78)	23.99 (2.60)	−2.52	0.008 *
22	22.99 (1.30)	21.46 (2.74)	−2.52	0.008 *
23	24.94 (3.11)	23.47 (2.33)	−2.52	0.008 *
31	20.69 (1.36)	20.38 (1.55)	−2.52	0.008 *
32	21.82 (1.21)	21.05 (1.17)	−2.52	0.008 *
33	23.98 (3.23)	22.14 (1.84)	−2.52	0.008 *
41	20.70 (2.63)	20.01 (1.35)	−2.52	0.008 *
42	22.71 (1.83)	20.25 (1.95)	−2.52	0.008 *
43	24.05 (4.23)	21.72 (2.83)	−2.52	0.008 *

Wilcoxon signed rank test; T1, before treatment; T2, At the end of levelling and alignment stage; # FDI notation; * significant difference (*p* < 0.05); IQR, interquartile range; *p*, *p*-value.

**Table 5 healthcare-11-00864-t005:** Root resorption between LLLT and non LLLT group, regardless of bracket system.

Variables ^#^	Median (IQR)		Z-Statistics	*p*
	LLLT (mm)	Non LLLT (mm)		
13	0.43 (1.08)	1.09 (1.73)	−0.49	0.635
12	0.26 (0.77)	0.49 (1.27)	−1.04	0.309
11	0.27 (0.55)	0.40 (0.62)	−1.19	0.242
21	0.29 (0.62)	0.56 (0.48)	−1.87	0.063
22	0.38 (0.87)	0.89 (1.46)	−2.17	0.029 *
23	0.49 (0.42)	1.02 (1.26)	−1.38	0.174
43	0.42 (0.65)	0.82 (1.39)	−1.89	0.060
42	0.63 (1.24)	0.62 (1.63)	−0.06	0.963
41	0.78 (0.98)	0.65 (1.01)	−0.69	0.497
31	0.29 (0.76)	0.26 (0.83)	−0.06	0.963
32	0.29 (0.69)	0.38 (0.57)	−0.78	0.450
33	0.92 (1.02)	0.99 (1.43)	−1.28	0.206

Mann Whitney test; # FDI notation; LLLT, Low-level laser therapy; * significant difference (*p* < 0.05); *p*, *p*-value; IQR, interquartile range; mm, millimetre.

**Table 6 healthcare-11-00864-t006:** Root resorption between CB and SL bracket system regardless of LLLT application.

Variables ^#^	Median (IQR)		Z-Statistics	*p*
	CB (mm)	SL (mm)		
13	0.39 (1.39)	0.85 (1.59)	−0.60	0.558
12	0.41 (0.77)	0.33 (1.29)	−0.07	0.948
11	0.28 (0.45)	0.34 (0.85)	−0.87	0.386
21	0.52 (0.79)	0.51 (0.34)	−0.43	0.676
22	0.62 (0.84)	0.55 (1.25)	−0.38	0.717
23	0.49 (0.85)	0.67 (0.90)	−1.05	0.300
43	0.72 (1.40)	0.45 (1.09)	−1.56	0.121
42	0.43 (1.14)	1.04 (1.68)	−0.98	0.336
41	0.83 (1.46)	0.61 (0.66)	−0.75	0.462
31	0.28 (0.64)	0.24 (0.87)	−0.38	0.717
32	0.44 (0.58)	0.27 (075)	−0.81	0.428
33	1.03 (1.66)	0.97 (0.81)	−0.69	0.497

Mann Whitney test; # FDI notation; *p*, *p*-value; IQR, interquartile range; mm, millimeter; CB, conventional bracket; SL, Self-ligating bracket.

**Table 7 healthcare-11-00864-t007:** Root resorption among all groups.

Tooth ^#^	Median (IQR)				*X*^2^ (df)	*p*
	CBNL	CBL	SLNL	SLL		
13	0.83 (1.37)	0.35 (1.52)	1.35 (3.46)	0.63 (0.96)	0.61 (3)	0.895
12	0.52 (0.74)	0.23 (0.80)	0.41 (1.72)	0.26 (0.75)	1.21 (3)	0.751
11	0.40 (0.56)	0.27 (0.34)	0.59 (1.29)	0.19 (0.74)	2.27 (3)	0.519
21	0.56 (0.46)	0.12 (0.94)	0.53 (0.62)	0.47 (0.56)	3.69 (3)	0.296
22	0.79 (0.72)	0.28 (0.85)	1.09 (1.87)	0.38 (0.94)	4.88 (3)	0.181
23	1.04 (1.31)	0.39 (0.30)	0.82 (1.68)	0.67 (0.69)	3.55 (3)	0.315
43	0.79 (1.35)	0.66 (1.57)	1.01 (1.60)	0.09 (0.48)	7.59 (3)	0.055
42	0.25 (1.50)	0.63 (1.13)	1.16 (1.68)	0.75 (2.00)	1.09 (3)	0.779
41	0.83 (1.53)	0.81 (1.52)	0.52 (0.65)	0.73 (0.81)	1.13 (3)	0.769
31	0.27 (0.98)	0.29 (0.50)	0.24 (0.79)	0.29 (1.53)	0.15 (3)	0.985
32	0.44 (0.55)	0.43 (0.60)	0.31 (0.81)	0.24 (0.73)	1.46 (3)	0.692
33	1.82 (1.58)	0.59 (1.13)	0.79 (0.98)	0.97 (0.99)	3.54 (3)	0.316

Kruskal Wallis test; # FDI notation; CBNL, conventional bracket non laser group; SLNL, self-ligating non laser group; SLL, self-ligating laser group; CBL, conventional bracket laser group; *p*, *p* value; X^2^, chi-square test; df, degree of freedom; IQR, interquartile range.

## Data Availability

Not applicable.

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
