# Peer review of "Effects of Low-Level Laser Therapy and Bracket Systems on Root Resorption during Orthodontic Treatment: A Randomized Clinical Trial"

_healthcare, 2023, doi:10.3390/healthcare11060864_

Round 1

Reviewer 1 Report

I have evaluated the whole manuscript:

Title: Effects of Low-Level Laser Therapy and Bracket Systems on Root Resorption During Orthodontic Treatment: a randomized clinical trial

By: Fazal Shahid, Shifat A Nowrin, Mohammad Khurssheed Alam, Mohd Fadhli Khamis, Adam Husein, Norma Ab Rahman *

This manuscript is a clinical trial that has been accompanied by approval ethics, and inform consent.

comments:

- Latest, there is a critical opinion on the benefits of using LLLT:

https://doi.org/10.3389/fbioe.2022.1089035

This will strengthen the introduction, especially for the statement "Laser Therapy has been used in Orthodontic Treatment for Many Years".

  - Because this is a clinical trial, the method needs to be shown a consort flow diagram.

- I think it will further increase readability if the author includes graphical abstract.

- Please include the 2010 Checklist Consort as the Supplementary File.

Author Response

Reviewer 1:

comments:

- Latest, there is a critical opinion on the benefits of using LLLT:

https://doi.org/10.3389/fbioe.2022.1089035

This will strengthen the introduction, especially for the statement "Laser Therapy has been used in Orthodontic Treatment for Many Years".

Reply to reviewer: Thank you so much for your comment. The reference has been added as per comment.

  - Because this is a clinical trial, the method needs to be shown a consort flow diagram.

Reply to reviewer: Thank you so much for your comment. Consort flow diagram has been added as per comment.

- I think it will further increase readability if the author includes graphical abstract.

Reply to reviewer: Thank you so much for your comment. We agree that a graphical abstract could increase the readability; however, we would like to keep the manuscript in traditional form.

- Please include the 2010 Checklist Consort as the Supplementary File.

Reply to reviewer: Thank you so much for your comment. Consort checklist has been added in supplementary file as per comment.

Reviewer 2 Report

Dear authors,

In this manuscript, authors assessed the outcomes of low-level laser therapy (LLLT) with the conventional bracket (CB) and self-ligating (SL) bracket systems on root resorption (RR) during orthodontic treatment. All the 24 patients were randomly divided into four individual groups (SLL: self-ligating laser, CBL: conventional bracket laser, SLNL: self-ligating non-laser, CBNL: conventional bracket non-laser). RR was measured from the cone-beam computed tomography (CBCT) radiographs which were taken at two stages of the orthodontic treatment: pre-treatment (T1) and after leveling and alignment stage (T2). Mann Whitney test compared the RR in laser and non-laser groups irrespective of the bracket systems and exhibited no significant differences except the left lateral incisor. Moreover, CB and SL groups showed no significant difference in RR among any tooth. Kruskal Wallis test was performed to compare the RR among all groups which presented no significant differences. They concluded that LLLT and bracket systems have no consequences on RR until the leveling and alignment stage of orthodontic treatment.

This manuscript has potential interest to evaluate the ability of laser devices to apply therapeutically.

Please check the below list.

1.Please describe how the irradiation conditions were determined.

2.Please add the kind of the laser machine used.

Author Response

Reviewer 2:

Dear authors,

In this manuscript, authors assessed the outcomes of low-level laser therapy (LLLT) with the conventional bracket (CB) and self-ligating (SL) bracket systems on root resorption (RR) during orthodontic treatment. All the 24 patients were randomly divided into four individual groups (SLL: self-ligating laser, CBL: conventional bracket laser, SLNL: self-ligating non-laser, CBNL: conventional bracket non-laser). RR was measured from the cone-beam computed tomography (CBCT) radiographs which were taken at two stages of the orthodontic treatment: pre-treatment (T1) and after leveling and alignment stage (T2). Mann Whitney test compared the RR in laser and non-laser groups irrespective of the bracket systems and exhibited no significant differences except the left lateral incisor. Moreover, CB and SL groups showed no significant difference in RR among any tooth. Kruskal Wallis test was performed to compare the RR among all groups which presented no significant differences. They concluded that LLLT and bracket systems have no consequences on RR until the leveling and alignment stage of orthodontic treatment.

This manuscript has potential interest to evaluate the ability of laser devices to apply therapeutically.

Please check the below list.

1.Please describe how the irradiation conditions were determined.

Reply to reviewer: Thank you so much for your comment. Information have been added in the methodology section.

2.Please add the kind of the laser machine used.

Reply to reviewer: Thank you so much for your comment. Details of the laser machine included in the methodology section.

Reviewer 3 Report

I reviewed the article “Effects of low-level laser therapy and bracket systems on root resorption during orthodontic treatment: A randomized clinical trial” and I believe that the paper shows some issues, mainly in the methodological procedure.

-        Line 40: specify EARR

-       Line 71: define moderate crowding;

-       Line 71: what is the rationale to perform CBCT scans? This is a fundamental question considering the aim of the study. No data about the acquisition parameters were reported.

-       Line 74: details about the sample size calculation and input data are missing. It’s not enough to write that it is done.

-       Line 98: the authors performed a CBCT scans “before starting the treatment (T1) and after finishing the leveling and alignment of the 98 treatment (T2).” Considering these two time points, there is a large radiation dose to the patient. Furthermore, in extraction therapies, what is the rationale for performing a second CBCT before starting the active phase of the work in which root resorption could occur?

-       Biomechanics used for space closure was not reported. It can be important for root resorption.

-       Line 106: there are no details on the operator and frequency of laser use.

-       Line 125: the paragraph of statistical analysis should be improved. Several details are missing.

-       A paragraph with study variables should be done: primary predictor variable and outcome variables should be better specified.

-       Study limitations should be revised and discussed better.

Author Response

Reviewer 3:

I reviewed the article “Effects of low-level laser therapy and bracket systems on root resorption during orthodontic treatment: A randomized clinical trial” and I believe that the paper shows some issues, mainly in the methodological procedure.

-        Line 40: specify EARR

Reply to reviewer: Thank you so much for your comment. The correction has been done as per comment.

-       Line 71: define moderate crowding;

Reply to reviewer: Thank you so much for your comment. The correction has been done as per comment.

-       Line 71: what is the rationale to perform CBCT scans? This is a fundamental question considering the aim of the study. No data about the acquisition parameters were reported.

Reply to reviewer: Thank you so much for your comment. The rationale to perform CBCT explained in the introduction section and acquisition parameters added in methodology section.

-       Line 74: details about the sample size calculation and input data are missing. It’s not enough to write that it is done.

Reply to reviewer: Thank you so much for your comment. The detail of the sample size calculation has been added.

-       Line 98: the authors performed a CBCT scans “before starting the treatment (T1) and after finishing the leveling and alignment of the 98 treatment (T2).” Considering these two time points, there is a large radiation dose to the patient. Furthermore, in extraction therapies, what is the rationale for performing a second CBCT before starting the active phase of the work in which root resorption could occur?

Reply to reviewer: Thank you so much for your comment. The reason of performing CBCT before starting the active phase has been explained in introduction section.

-       Biomechanics used for space closure was not reported. It can be important for root resorption.

Reply to reviewer: Thank you so much for your comment. The current study was conducted until the leveling and alignment stage of the orthodontic treatment. Space closure were not a part of this study.

-       Line 106: there are no details on the operator and frequency of laser use.

Reply to reviewer: Thank you so much for your comment. Details of the laser machine included in the methodology section.

-       Line 125: the paragraph of statistical analysis should be improved. Several details are missing.

Reply to reviewer: Thank you so much for your comment. Information have been added as per comment.

-       A paragraph with study variables should be done: primary predictor variable and outcome variables should be better specified.

Reply to reviewer: Thank you so much for your comment. Correction has been done as per comment.

-       Study limitations should be revised and discussed better.

Reply to reviewer: Thank you so much for your comment. Limitation part has been revised in discussion section as per comment.

Reviewer 4 Report

Dear Authors,

Please find below some observations and recommendations concerning your article entitled” Effects of low-level laser therapy and bracket systems on root resorption during orthodontic treatment: A randomized clinical trial”.

Title

Please follow the MDPI authors' guidelines.

In the Abstract section:

- Please follow the MDPI authors' guidelines concerning the abstract structure (no more than 200 words should be included, without headings).

- Please rewrite the abstract section

- Keywords: Please write them properly, not only the abbreviations.

In the Introduction section:

- Please include the null hypothesis at the end of the introduction sections

In the Materials and Methods section:

- Please include information regarding the trial and patient recruitment periods.

-line 72- ”This randomized clinical trial followed the CONSORT guidelines”, please add the flow diagram.

Sample size calculation:

-line 74- Please add details about software used, version, Manufacturer, City and State.

Data collection procedure:

-lines 78-80- please add details about brackets: commercial name, manufacturer, City and State

- line 82- please add details about” buccal tubes (3M Unitek)”: manufacturer, City and State

- line 84- please add details about” nickel-titanium (NiTi) archwire of 3M Unitek”: manufacturer, City and State

            -please add details about” A Ga-Al-As diode laser (Ilase, USA)” manufacturer, City

- line 86- please add details about” Planmeca RomexisTM software”: manufacturer, City and State

- lines 88-89- ” Based on the inclusion criteria, declined participation, and other reasons....” please provide more information on the excluding criteria of the patients from the study group.

- line 102 Please add how LLLT was applied: Time (day/week), number of laser applications, place buccal/palatal/lingual mucosal surface

- lines 104-106 should be moved after the new paragraph.

Statistical analysis

-line 125- Please add details about software used, version, Manufacturer, City and State.

In the Discussion section:

- Please add one paragraph about the null hypothesis.

- Please add the limitations of this study

In the References section: please follow the styles recommended for MDPI journals.

Author Response

Reviewer report:

Reviewer 4:

Dear Authors,

Please find below some observations and recommendations concerning your article entitled” Effects of low-level laser therapy and bracket systems on root resorption during orthodontic treatment: A randomized clinical trial”.

Title

Please follow the MDPI authors' guidelines.

Reply to reviewer: Thank you so much for your comment. The title of the study followed the MDPI guideline.

In the Abstract section:

- Please follow the MDPI authors' guidelines concerning the abstract structure (no more than 200 words should be included, without headings).

Reply to reviewer: Thank you so much for your comment. The abstract contains 199 words without any heading.

- Please rewrite the abstract section

Reply to reviewer: Thank you so much for your comment. We were unable to comprehend which exact part of the abstract need to be changed as the current abstract is representing the summery of the study.

- Keywords: Please write them properly, not only the abbreviations.

Reply to reviewer: Thank you so much for your comment. Keywords have been corrected as per comment.

In the Introduction section:

- Please include the null hypothesis at the end of the introduction sections

Reply to reviewer: Thank you so much for your comment. Null hypothesis has been added as per comment.

In the Materials and Methods section:

- Please include information regarding the trial and patient recruitment periods.

Reply to reviewer: Thank you so much for your comment. Trial period was mentioned as per comment.

-line 72- ”This randomized clinical trial followed the CONSORT guidelines”, please add the flow diagram.

Reply to reviewer: Thank you so much for your comment. Flow diagram has been added as per comment.

Sample size calculation:

-line 74- Please add details about software used, version, Manufacturer, City and State.

Reply to reviewer: Thank you so much for your comment. The details have been added.

Data collection procedure:

-lines 78-80- please add details about brackets: commercial name, manufacturer, City and State

Reply to reviewer: Thank you so much for your comment. Details have been added as per comment.

- line 82- please add details about” buccal tubes (3M Unitek)”: manufacturer, City and State

Reply to reviewer: Thank you so much for your comment. Details have been added as per comment.

- line 84- please add details about” nickel-titanium (NiTi) archwire of 3M Unitek”: manufacturer, City and State

Reply to reviewer: Thank you so much for your comment. Details have been added as per comment.

-please add details about” A Ga-Al-As diode laser (Ilase, USA)” manufacturer, City

Reply to reviewer: Thank you so much for your comment. Details have been added as per comment.

- line 86- please add details about” Planmeca RomexisTM software”: manufacturer, City and State

Reply to reviewer: Thank you so much for your comment. Details have been added as per comment.

- lines 88-89- ” Based on the inclusion criteria, declined participation, and other reasons....” please provide more information on the excluding criteria of the patients from the study group.

Reply to reviewer: Thank you so much for your comment. Details have been added as per comment.

- line 102 Please add how LLLT was applied: Time (day/week), number of laser applications, place buccal/palatal/lingual mucosal surface

Reply to reviewer: Thank you so much for your comment. Details have been added in data collection procedure section as per comment.

- lines 104-106 should be moved after the new paragraph.

Reply to reviewer: Thank you so much for your comment. Correction has been done as per comment.

Statistical analysis

-line 125- Please add details about software used, version, Manufacturer, City and State.

Reply to reviewer: Thank you so much for your comment. Information have been added as per comment.

In the Discussion section:

- Please add one paragraph about the null hypothesis.

Reply to reviewer: Thank you so much for your comment. Correction has been done as per comment.

- Please add the limitations of this study

Reply to reviewer: Thank you so much for your comment. Limitations have been added as per comment.

In the References section: please follow the styles recommended for MDPI journals.

Reply to reviewer: Thank you so much for your comment. References followed the MDPI journal style.

Round 2

Reviewer 4 Report

Dear authors,

Thank you very much for revising the manuscript according to my comments.